# Patterns of Differentially Expressed circRNAs in Human Thymocytes

**DOI:** 10.3390/ncrna8020026

**Published:** 2022-03-30

**Authors:** Pilar López-Nieva, Pablo Fernández-Navarro, María Ángeles Cobos-Fernández, Iria González-Vasconcellos, Raúl Sánchez Pérez, Ángel Aroca, José Fernández-Piqueras, Javier Santos

**Affiliations:** 1Genome Dynamics and Function Program, Genome Decoding Department, Severo Ochoa Molecular Biology Center (CBMSO), CSIC-Madrid Autonomous University, 28049 Madrid, Spain; macobos@cbm.csic.es (M.Á.C.-F.); iria.gonzalez@cbm.csic.es (I.G.-V.); jfpiqueras@cbm.csic.es (J.F.-P.); jsantos@cbm.csic.es (J.S.); 2Institute of Health Research Jiménez Díaz Foundation, 28040 Madrid, Spain; 3Cancer and Environmental Epidemiology Unit, National Center for Epidemiology, Institute of Health Carlos III, 28029 Madrid, Spain; 4Consortium for Biomedical Research in Epidemiology and Public Health (CIBERESP), 28029 Madrid, Spain; 5Department of Congenital Cardiac Surgery, Hospital Universitario La Paz, 28046 Madrid, Spain; raulcaravaca@hotmail.com (R.S.P.); aarocap@telefonica.net (Á.A.)

**Keywords:** human thymocytes, circRNAs, circRNA–miRNA–mRNA controlling networks

## Abstract

Circular RNAs (circRNAs) are suggested to play a discriminative role between some stages of thymocyte differentiation. However, differential aspects of the stage of mature single-positive thymocytes remain to be explored. The purpose of this study is to investigate the differential expression pattern of circRNAs in three different development stages of human thymocytes, including mature single-positive cells, and perform predictions in silico regarding the ability of specific circRNAs when controlling the expression of genes involved in thymocyte differentiation. We isolate human thymocytes at three different stages of intrathymic differentiation and determine the expression of circRNAs and mRNA by RNASeq. We show that the differential expression pattern of 50 specific circRNAs serves to discriminate between the three human thymocyte populations. Interestingly, the downregulation of *RAG2*, a gene involved in T-cell differentiation in the thymus, could be simultaneously controlled by the downregulation of two circRNASs (hsa_circ_0031584 and hsa_circ_0019079) through the hypothetical liberation of hsa-miR-609. Our study provides, for the first time, significant insights into the usefulness of circRNAs in discriminating between different stages of thymocyte differentiation and provides new potential circRNA–miRNA–mRNA networks capable of controlling the expression of genes involved in T-cell differentiation in the thymus.

## 1. Introduction

Circular RNAs (circRNAs) are stable single-stranded RNA molecules that form a covalently closed continuous loop. Based on the source of the genome and biogenesis patterns, circRNAs are mainly divided into three groups: exonic circRNAs (EcircRNAs), exonic-intronic circRNAs (EIciRNAs) and circular intronic RNAs (ciRNAs) [1,2,3]. They are involved in multiple biological processes, functioning mainly through interactions with microRNAs and proteins or by the expression of specific peptides [2,4,5,6]. Although details of known circRNA-effector mechanisms and functions in physiology and many human diseases are described in recent reviews [7,8,9,10,11], our understanding of the different roles in normal physiological conditions is limited for the vast majority of identified circRNAs [12]. A better understanding of the mechanisms underlying the generation and functions of circRNAs in the maturational processes of human intrathymic thymocytes will help us understand the physiological and pathological processes to which they are subjected and prepare the pipeline for circRNA-based therapeutic intervention and the diagnosis of human hematological diseases in the future. Preliminary data exist on circRNA expression in different peripheral blood cell populations [13,14]. The deregulation of circRNAs is shown to be key in the development of T-cell acute lymphoblastic leukemia (T-ALL) and serves to discriminate between some populations of normal thymocytes [15] but not all. This study aims to investigate the expression patterns of differentially expressed circRNAs in three different development stages of human intrathymic thymocytes. Together with this, in silico predictions are performed to determine the ability of differentially expressed circRNAs in controlling the expression of coding genes involved in thymocyte differentiation.

## 2. Results and Discussion

### 2.1. Isolation of Human Thymocytes at Different Stages during Their Intrathymic Differentiation

As circRNAs are known to be expressed in tissue specifically [16], it should be no surprise that circRNAs are also involved in this specific developmental process. However, not much is known about circRNAs involved in human thymopoiesis. Normal thymopoiesis is a strictly regulated developmental process that is initiated by early T-cell progenitors (CD34+) migrating from the bone marrow into the thymus. Within this microenvironment, distinct stages of T-cell development can be identified by a combination of cell surface markers (CD34, CD4, CD8, CD3, etc.), and each of these stages contains a distinct transcriptional profile [17,18,19,20,21,22,23]. On this premise, human postnatal thymocytes were isolated from thymuses removed during the corrective congenital cardiac surgery of seven pediatric patients aged between 1 month and 4 years old; none of the patients recruited for this study had any chromosomal abnormalities, oncologic processes or genetic conditions with a propensity to develop them. Intrathymic thymocytes were divided in three main stages: early immature CD34+CD2− (ST1, Stage 1; 100% double negatives CD4−CD8−); intermediate CD1A+ (ST2, Stage 2; 88% double-positive thymocytes CD4+CD8+); and mature CD1A− thymocytes (ST3, Stage 3; 95% single positive thymocytes CD4+CD8− and CD4−CD8+) (see Section 3 and Appendix A for details).

### 2.2. circRNAs Are Differentially Expressed during Thymocyte Differentiation

A total of 1004 circRNAs were detected in the nine samples analyzed (three replicates for each of the three thymocyte populations), the vast majority being of the exon–exon type (96.3%). To assess whether the expression levels of these circRNAs could be a useful tool to differentiate thymocyte populations, we performed a hierarchical cluster analysis of the sample expression profiling using the full linkage method to estimate the Euclidean distance using the R package NbClust. The samples were segregated into three clusters corresponding to the three different populations of thymocytes (Figure 1A). An unsupervised principal component analysis (PCA analysis) was also performed that separated thymocyte populations, in the same way, pointing towards prominent differences in circular RNA expression among them (Figure 1B,C). Pairwise comparisons between the circRNAs of the three thymocyte populations revealed 50 circRNAs differentially expressed in at least one pairwise comparison (ST2 vs. ST1, ST3 vs. ST1 and ST3 vs. ST2) (Figure 2A, Appendix A); none of which are found in the databases known to be associated with thymus differentiation. The number of differentially expressed circRNAs between the three thymocyte populations are shown in a Venn diagram providing insight into the similarities and differences (Figure 2B). Strikingly only the hsa_NEIL3_0001 was found to be expressed in the three comparison sets. The hierarchical cluster analysis of sample expression profiles (z-score) using only the 50 differentially expressed circRNAs also separated the samples into the three populations of thymocytes (Figure 2C).

### 2.3. Circular-to-Linear Expression Proportion

Interestingly, the circular-to-linear expression proportion (CLP), the proportion of expression between the circular and linear isoforms, revealed that circular RNAs are expressed less abundantly than their respective linear counterparts, with the exception of hsa_RP11-563D10_0001, which exhibits the same number of circular and linear counts in ST1 thymocytes (CLP: 0.5) (Appendix A). Differences in CLP values for many circRNAs suggest a certain degree of independence in the expression control of linear and circRNAs (Figure 3). In any case, it seems that the function of the circRNAs is different from that of their linear partners, as is evident in the case of HIPK3 [24].

### 2.4. Validation of Selected circRNAs

To verify the reliability of the RNA-seq data, seven circRNAs were randomly selected for further validation experiments. Convergent (for linear transcripts) and divergent (for circular RNAs) specific primers were designed for RT-qPCR (Appendix A) to verify the differential expression seen in the RNA-seq data. Quantification by qRT-PCR in ST1, ST2 and ST3 populations confirmed RNA-seq results for all tested mRNAs and circRNAs, supporting data robustness and reproducibility. Significant deregulation of circRNAs was confirmed (Figure 4). The comparison between the fold change data between the different populations (ST1-ST2/ST1-ST3/ST2-ST3) and the qPCR data obtained when we performed a multiple comparisons analysis (ANOVA) shows that there is a sufficiently robust correlation between both methodologies (Appendix A). The PCR products were visualized using a 2% Ethidium Bromide agarose gel followed by band purification. Sanger sequencing was performed to validate the PCR products, and the circRNA junctions were identified unambiguously (Appendix A).

### 2.5. Patterns of Differentially Expressed circRNAs

The expression patterns of the differentially expressed circRNAs (z-scores profiles by populations of thymocytes) identified eight cohorts or clusters of circRNAs (Figure 5A), which are shown in the boxplots of Figure 5B. Cluster 1 consist of hsa_circIKZF1_0001, which shows a progressive increase in expression as thymocyte differentiation progresses (Appendix A). This circRNA is a T-cell specific circRNA in mature blood cell populations [14]. Cluster 3 includes four circRNAs, hsa_circHIPK3_0001 being one of them. This cluster presents a pattern opposite to the previous one, in which expression decreases as thymocyte differentiation progresses. Interestingly, a recent work reported that circHIPK3, but not HIPK3 mRNA, could serve as a modulator of cell growth and cell proliferation in different human cells by sponging multiple miRNAs in human cells [24]. Cluster 6 is a heterogeneous one that comprises 15 circRNAs, including hsa_circLEF1_0001, which is significantly upregulated in the ST2 when compared to the other two stages (Appendix A; Average Counts: ST1: 3; ST2: 48; ST3: 12).

### 2.6. mRNAs Differentially Expressed during Thymocyte Differentiation

Since differentially expressed circRNAs could be controlling the expression of mRNAs by acting as sponges for specific microRNAs, we determined the mRNA expression profiles of the purified thymocyte populations at three different stages of maturation (SP1, SP2 and SP3). A total of 17,804 mRNAs were detected. Of them, 5103 were differentially expressed mRNAs in any of the comparisons made (|log2FC| more than or equal to 1 and *p*-value adjusted by a BH procedure (p.adjusted) score less than or equal to 0.05) (Appendix A). Notably, 180 of these genes were involved in T-cell differentiation. Our results indicated that each thymic population was characterized by a distinct mRNA expression pattern, which reflected the developmental relationships across maturation stages in T precursors. Differentially expressed mRNAs among the three thymocyte populations are shown in Appendix A (see Go Ontology column in Appendix A).

### 2.7. In Silico Functional Outcome Prediction of Specific circRNAs Differential Expression

To identify potentially functional circRNA–miRNA–mRNA regulatory networks (Appendix A), we first predicted miRNA binding sites in circRNA sequences using CircInteractome. Following, we identified miRNA–mRNA interactions using TargetScan as indicated in the Section 3. A total of 1035 miRNA binding sites were predicted in the 35 circRNA sequences that were identified in mirBase (available online: https://www.mirbase.org/ (accessed on 23 January 2021)) [25] out of the 50 circRNAs differentially expressed in this study. Only 206 miRNAs binding sites showed a “context score percentile” more than 95, identifying a total of 127 different miRNAs. A total of 285 target genes were identified for these miRNAs using TargetScan with a Cumulative.weighted.context score less than or equal to −1. Only 66 of these genes were differentially expressed in any of the comparisons made. Interestingly, the expression patterns of these mRNAs (z-scores profiles by populations of thymocytes) serve to discriminate between the three stages of thymocyte differentiation (Figure 6). Finally, a total of 95 networks were constructed, merging circRNA, miRNAs and the selected genes (Appendix A). Of all these networks, 38 included a circRNA very possibly acting as a miRNA sponge, 12 included a circRNA possibly acting as a miRNA sponge and 45 did not include a circRNA acting as a miRNA sponge, according to the criteria established and described above (Figure 7).

Interestingly, the downregulation of the *RAG2* gene (which encodes a protein involved in the initiation of V(D)J recombination during T cell development) from ST1 to ST3 stages could be simultaneously controlled by the downregulation of hsa_circ_0031584 (expressed by *ARHGAP5 R* gene) and hsa_circ_0019079 (expressed by *KIF20B* gene) through the hypothetical liberation of hsa-miR-609. Further experimental approaches would eventually confirm the involvement of circRNAs in controlling genes directly involved in T-cell differentiation in the thymus.

## 3. Materials and Methods

### 3.1. Patients’ Characteristics

The ages at the time of surgery were 1 week (1), 2 weeks (2), 4 weeks (1), 14 weeks (1), 20 weeks (1) and 4 years and two months (1). Five of them were male and two were female; two of the patients had Shone’s complex (a rare congenital cardiac malformation characterized by a complex of four obstructive lesions in the left heart) and Tetralogy of Fallot (is an obstruction of the pulmonary outflow tract, a ventricular septal defect (VSD) due to misalignment, an overriding aorta and right ventricular hypertrophy) (for more information see Appendix A). The fact that there is no known association between these pathologies and abnormalities in thymus development allowed us to include them in the study. To determine whether segregation CD4/CD8 ratios were comparable between different human samples at the ages used in this work, we performed a flow cytometric study of these samples. With the results obtained and taking into account the intra-individual differences, we can assure that the population profiles of human cells from pediatric thymuses used in this study have very similar population profiles and their immunophenotypes were completely normal (Appendix A). In all cases, the informed consent of the operation was signed by the legal representatives of the children in accordance with the Declaration of Helsinki, in which it was specified that as a consequence of the operation, the thymectomy would be performed. Institutional review board approval was obtained for these studies (CEI 98-1825).

No deaths were recorded during the survey period. None of the patients required intensive care because of immunologic complications, including acute or recurrent infectious diseases, such as bacteremia and mediastinitis.

### 3.2. Isolation of Human Thymocytes at Different Stages of Differentiation

To isolate early immature thymocytes (ST1), cell suspensions were firstly enriched in CD2− thymocytes using the sheep red blood cell technique [26,27]. From this population, CD34+ cells were isolated with appropriate Ig-coated magnetic-activated MicroBeads using autoMACS Pro (Miltenyi Biotec, Bergisch Gladbach, Germany). On the other hand, starting from non-manipulated thymocytes, the CD1A+ (ST2) population was isolated with CD1A+ MicroBeads using autoMACS Pro and its immunophenotype CD4+CD8+ was determined by flow cytometry. From the remaining CD1A− cell population, we selected the ST3 stage that was sorted by possessing either CD4+ or CD8+ single-positive thymocytes using flow cytometry. All the immunophenotypes of the thymocyte populations were magnetically sorted and validated by flow cytometry using a FACSCalibur cytometer (Becton Dickinson, San Diego, CA, USA) with the following antibodies: CD34-PE, CD8-APC and CD4-PE (all from MACS Miltenyi Biotec), revealing more than 98% purification efficiency. In order to have enough cells to carry out the studies, once they were fractionated, they were pooled to obtain the ST1, ST2 and ST3 populations with which we worked.

### 3.3. RNA Isolation

Total RNA was obtained using TriPure Reagent (Roche Applied Science, Indianapolis, IN, USA), following the manufacturer’s instructions. RNA integrity numbers (RIN) were in the range of 7.2–9.8. Image analysis, per-cycle basecalling and quality score assignment were performed with Illumina Real-Time Analysis software (Illumina, San Diego, CA, USA).

### 3.4. Quality Control and Trimming

High-quality RNA samples were used for high-depth Illumina total RNA sequencing (RNA-seq) after ribosomal depletion with 3 replicates for each thymocyte population (NIMGenetics and Helix BioS, Scientific Park of Madrid, Madrid, Spain). The RNA detection and expression workflow carried out in this study allowed us to discover differentially expressed circular RNAs and their linear counterparts (host genes). The quality control was carried out with FastQC and Fastp [28]. From this stage, two pipelines were run to obtain the mRNA expression arrays and the detection and quantification of circRNAs together with their linear counterparts.

### 3.5. mRNA Pipeline

The aim of this step was to align the processed RNA-seq reads against the reference genome using the HISAT2 alignment tool [29]. For the alignment, we used GRCh37/hg19, Ensembl version 87. In this stage, the assembly of transcripts from which the transcriptional expression of the samples can be estimated was carried out. Such expression was performed using the StringTie suite [30]. This is a highly efficient assembler designed to align RNA-seq data using a network flow algorithm. At the same time, it assembles and quantifies expression levels for transcriptome features in a readable Ballgown-like format (via the -B option). Expression pattern analysis makes it possible to identify and extract cohorts of genes that behave in a coordinated manner with respect to the complete set of genes and associate it with a particular biological context [31,32,33,34]. This type of analysis provides evidence of possible gene and/or functional interactions between genes co-expressed under different conditions or over time.

### 3.6. circRNA Pipeline

The sequences that did not exceed the Q30 score and read less than 100 base pairs in length were eliminated. The alignment of RNA-seq reads was established against the reference genome (GRCh37/hg19, Ensembl version 87) and was performed with the STAR alignment tool [35]. To carry out the initial exploratory analysis, we started from the normalized expression matrices using the variance stabilizing transformation algorithm (vst) of DESeq2 [36] and applying a local adjustment for circular and linear RNAs, taking into account the variance in each case. FASTQ data were deposited in the NCBI Gene Expression Omnibus and are accessible through GEO accession number GSE178889.

### 3.7. Quantification and Annotation of circRNAs

The quantification and annotation of circular RNAs and their linear counterparts was performed using circTools [37] since it is one of the few bioinformatics tools that allows the alignment of the sequences mates separately, allowing the detection of exonic, exonic-intronic and intergenic circRNAs and helping to perform internal controls on the sample itself for better identification of chimeric junctions.

### 3.8. Pairwise Comparations of circRNA and mRNA Expression among the Three Thymocyte Populations

Pairwise comparations of circRNAs and mRNA expression among the three thymocyte populations were carried out using the Wald statistic. For multiple comparisons, the *p*-value was adjusted using the Benjamini–Hochberg (BH) procedure. We considered significant differential expression when log2FC was greater than or equal 1 (in absolute numbers) and when the *p*-value of contrasts adjusted using BH was less than 0.05. The circular-to-linear expression proportion (CLP) was adapted from that provided by the R package CircTest [38].

Differential expression analysis was performed using the DESeq2 package [36]. Other parameters applied were an internal independent filter with a local model fit for circRNA and a parametric fit for mRNA and a normalization ratio with the replacement of default outliers, including the developmental stages of control thymocytes as a factor.

### 3.9. Functional Annotation of circRNAs

Several specific databases for circular RNAs were consulted, including CircFunBase [39], which uses data from Circular RNA Interactome, circBase, CIRCpedia, among others and the circAtlas 2.0 database [40]. We used the nomenclature of the circAtlas 2.0 database (Available at: http://circatlas.biols.ac.cn (accessed on 20 July 2021)), although the correspondence with the nomenclatures of other databases is available in Appendix A.

### 3.10. mRNA-circRNA-miRNA Interaction Network Analysis

To visualize mRNA–circRNA–miRNA interaction network analysis, we used Cytoscape 3.9.1 software (Available at: https://www.cytoscape.org (accessed on 10 February 2022)) [41], a tool for identifying molecular interaction networks and biological pathways and integrating these networks with annotations, gene expression profiles and other status data. Within the tools menu, we used a section dedicated to Merge networks. Using its editing tools, we identified mRNA, circRNA and miRNA with different colors and icons.

### 3.11. Retrotranscription and Polymerase Chain Reactions

Special divergent were designed for each circRNA. DNA was amplified using Immolase Taq polymerase (Bioline USA Inc., Kenilworth, NJ, USA). The reaction parameters were: 95 °C for 8 min; followed by 40 cycles of 95 °C for 1 min, an appropriate annealing temperature (according to the melting temperature of the primers) for 1 min and 72 °C for 2 min; and 72 °C for 10 min. The resulting PCR products were gel-purified (2% agarose electrophoresis with ethidium bromide (EtBr)) with Wizard^®^ SV Gel and a PCR Clean-Up System (Promega, Madison, WI, USA). Sanger DNA sequencing of the PCR-amplified fragments (in both directions) was performed by a Macrogen Europe sequencer (Amsterdam, The Netherlands).

Special divergent and convergent primers were designed to verify the reliability of RNA-seq data, and cDNA was synthesized with a High Capacity RNA-to-cDNA Kit (Applied Biosystems, Waltham, MA, USA). qPCR was performed on an ABI 7500 Real-time PCR Detection System (ABI, Los Angeles, CA, USA). The housekeeping genes *B2M* and *PPIA* were used as an internal control. The data were analyzed using the 2^−ΔΔCt^ method and presented as relative expression levels from three biological replicates and three parallel technical replicates. All detailed PCR conditions and primers sequences are listed in Appendix A.

### 3.12. CircRNA Functional Predictions

MiRNA binding sites were predicted in circRNA sequences using the web tool Circular RNA Interactome [42]. MiRNAs with a context score percentile more than 95 were selected, and miRNA target genes were retrieved with TargetScan [43]. Genes with a Cumulative.weighted.context score less than or equal to −1 were selected. Then, the results from the mRNA expression analyses were used to construct the final circRNA–miRNA gene networks. Only those networks that included genes that were differentially expressed in any of the analyzed comparisons were considered. Finally, to assess which network included circRNAs that possibly control the miRNA-targeted gene expression through sponging the miRNAs, the log2 of fold change and FDR values of the expression contrasts performed were taken into account. In that way, we classified networks as (a) “strongly possible sponge” (Type = 2) (network including a circRNA very possibly acting as miRNA sponge), when in any of the three comparisons performed, the log2 of fold change observed in both the circRNA and mRNA analyses were greater than or equal to 1 in absolute value, had the same sign in both contrasts and FDR values were less than or equal to 0.05; (b) “possible sponge” (type = 1) (network including a circRNA possibly acting as miRNA sponge) when in any of the three comparisons performed, the log2 of fold change observed in both the circRNA and mRNA analyses were greater than or equal to 1 in absolute value and had the same sign. (c) “no sponge” (type = 0) (network not including a circRNA acting as miRNA sponge), for any other option.

## 4. Conclusions

In line with previous reports using only less mature fractions [15], these results show, for the first time, the usefulness of the circRNAs to discriminate between three different stages of thymocyte differentiation and provides new potential circRNA–miRNA–mRNA networks capable of controlling the expression of genes involved in T-cell differentiation in the thymus. These results lead us to believe that circRNAs could have, if appropriately modified, a potential role to act as molecular or therapeutic tools to regulate cellular stability through their interaction with miRNAs and other RNAs or RNA-binding proteins.

## Figures and Tables

**Figure 1 ncrna-08-00026-f001:**
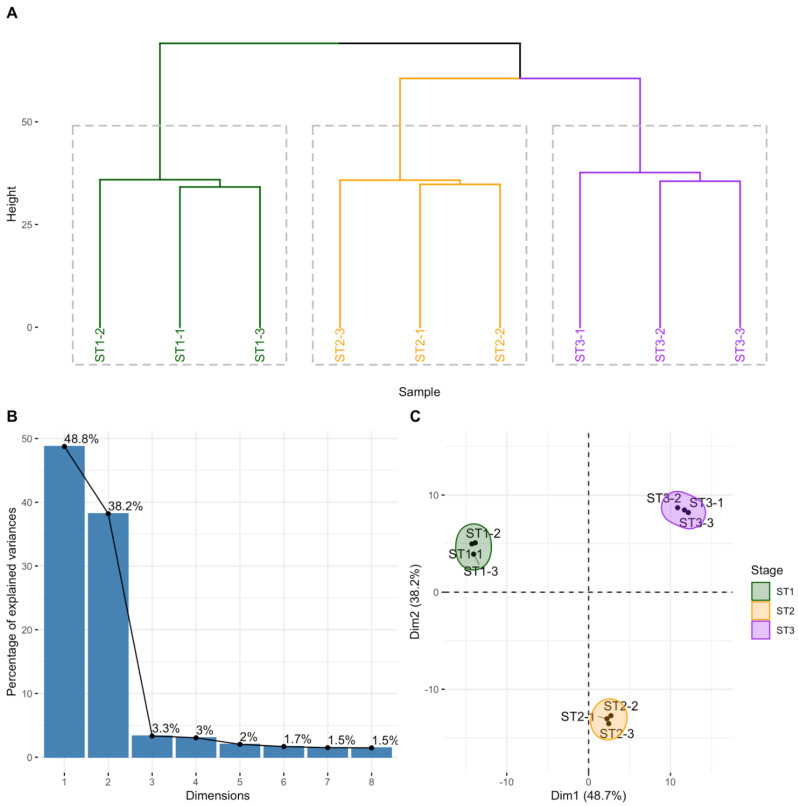
Results of the cluster and principal component analyses. (**A**) Dendrogram of the cluster analysis solution; (**B**) Screen plot of the percentage of sample expression variability explained by the principal components/dimensions identified in the principal component analysis (PCA); (**C**) Graph of samples scores in the first two dimensions/principal components of the PCA analysis. The figures are colored in the replicas according to the three levels that make up the stages of thymocyte maturation (ST1-3).

**Figure 2 ncrna-08-00026-f002:**
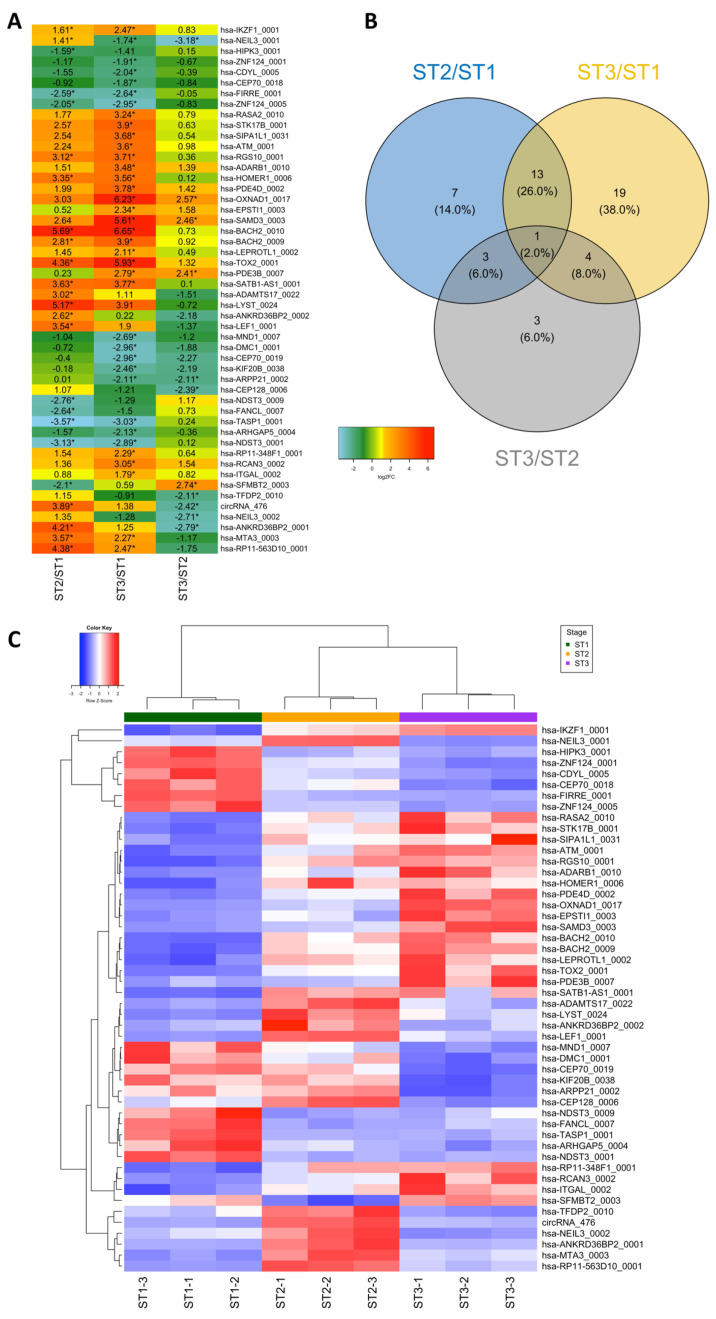
Pairwise comparison analysis and differential expression of circRNAs. (**A**) Heatmap representation depicting changes in the expression of the 50 circRNAs differentially expressed in at least one pairwise comparison. The numbers in each box represent the fold changes (log2FC) calculated as the ratio of the read counts between the groups of samples compared. Color key interpretation is indicated in the upper center part of the figure. Asterisks indicate a value that fulfills the two criteria for significant differential expression; (**B**) Venn diagram depicting the overlap of the 50 circRNAs differentially expressed in pairwise comparisons between the three thymocyte populations; (**C**) Heatmap representing the expression of the 50 circRNAs differentially regulated in the thymocytes at the three stages of intrathymic differentiation in each sample. Expression level (z-score) is represented as a colored cell. The color of each of these cells depends on the circRNA expression level. A tricolor scale is used: red color represents high expression, white represents medium expression and blue color represents low expression. Color key interpretation is indicated in the upper left part of the figure.

**Figure 3 ncrna-08-00026-f003:**
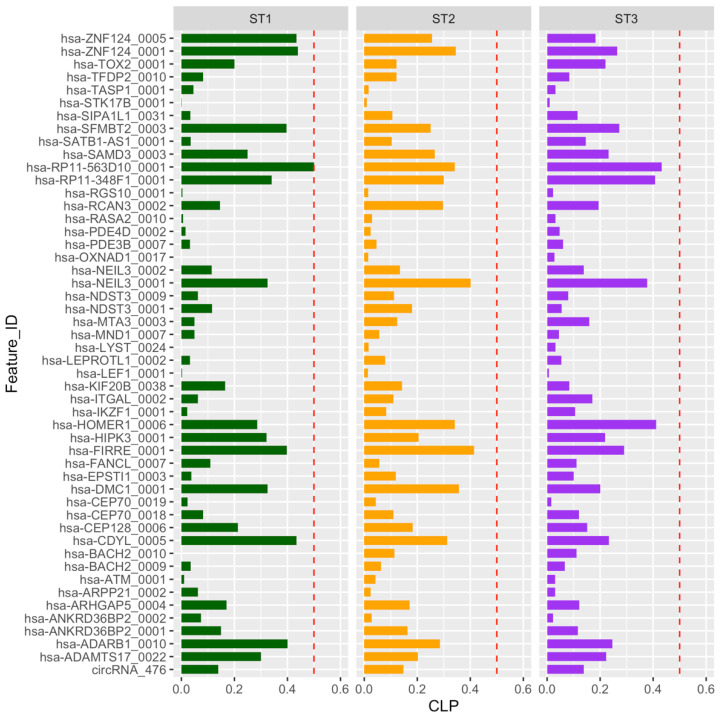
Circular-to-linear expression proportion (CLP) for the circRNAs differentially expressed during thymocyte differentiation in each stage (ST). Red line marks a CLP value of 0.5.

**Figure 4 ncrna-08-00026-f004:**
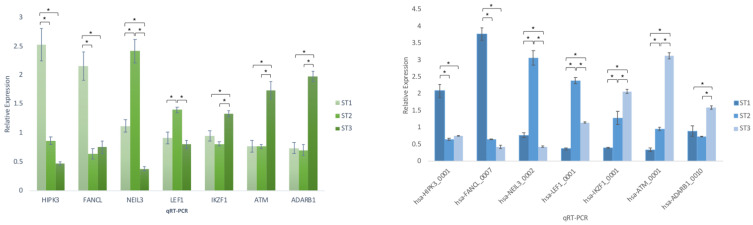
qPCR validation of linears (mRNAs) and their counterparts circRNAs. Quantitative data are presented as the mean ± standard deviation (SD). One-way ANOVA analysis and paired Student’s *t*-test were used to assess significance among groups. Following one-way ANOVA, Tukey’s post hoc test was performed. SPSS software (v27; SPSS, Inc., Chicago, IL, USA) was used to process all statistical analyses. All tests were performed in triplicate (*n* = 3). A *p*-value < 0.05 was considered to indicate a statistically significant difference. * Denotes a statistical significance.

**Figure 5 ncrna-08-00026-f005:**
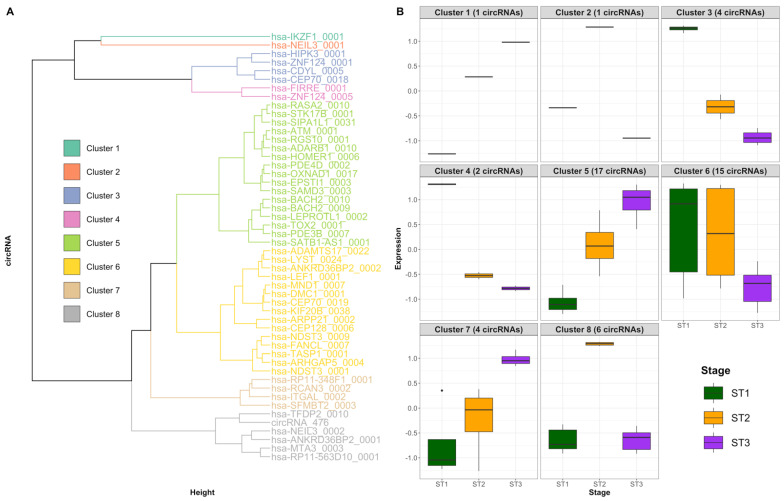
Cluster sample and circRNA analysis using the 50 circRNAs differentially expressed in at least one pairwise comparison. (**A**) Dendrogram of the solution from the hierarchical cluster analysis of the expression profiles of the 50 circRNAs differentially expressed in pairwise comparisons between the three thymocyte populations. (**B**) Boxplot showing the z-scores of the thymocyte populations in each of the 8 clusters of circRNA identified in this study.

**Figure 6 ncrna-08-00026-f006:**
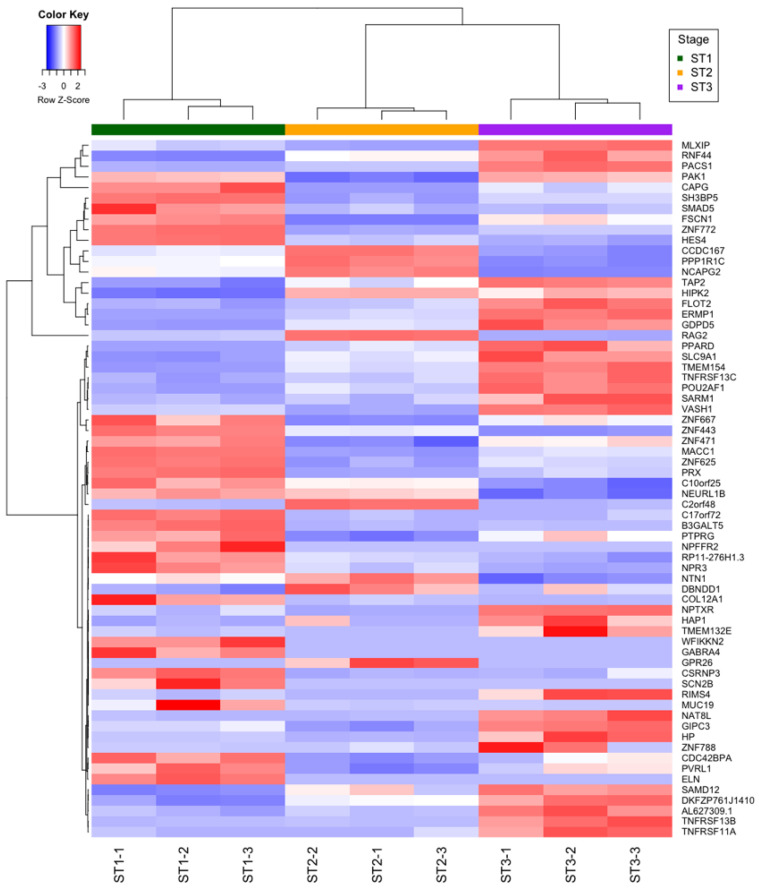
Expression patterns of the differentially expressed mRNAs (z-scores profiles by populations of thymocytes), According to all the parameters used during this work, we propose 3 as the most appropriate number of clusters of identified mRNAs.

**Figure 7 ncrna-08-00026-f007:**
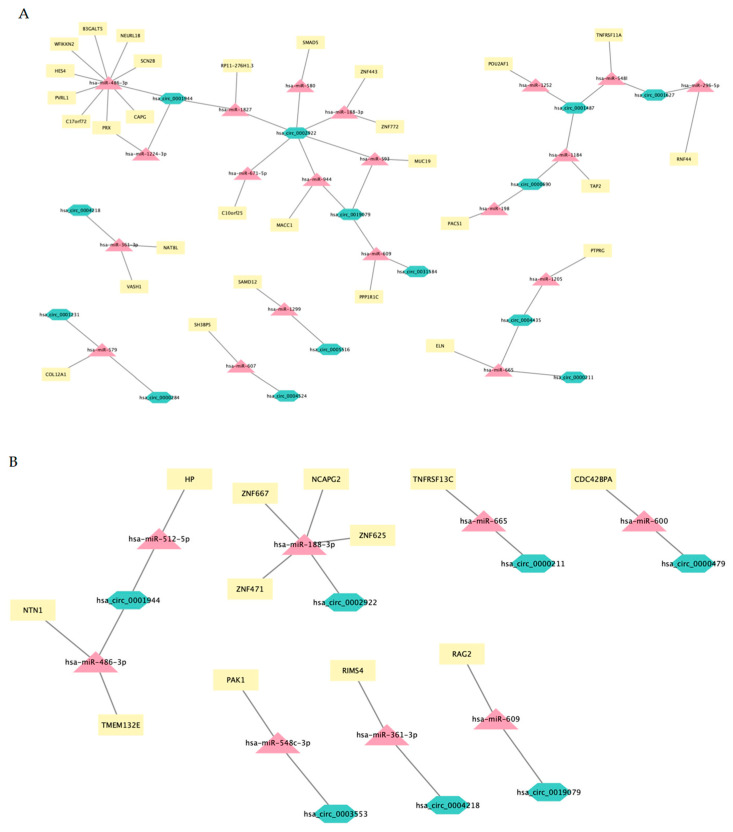
mRNA–circRNA–miRNA networks. Networks including circRNAs “very possibly” acting as miRNA sponge (Type 2) (**A**), “possibly” acting as miRNA sponge (Type 1) (**B**) and “not” acting as miRNA sponge (Type 0) (**C**). The network type 2 consists of 14 circRNAs, 20 miRNAs and 29 genes were generated by Cytoscape 3.9.1. The network Type 1consists of 7 circRNAs, 8 miRNAs and 12 genes and was generated by Cytoscape 3.9.1. The Network Type 0, consisting of 18 circRNAs, 26 miRNAs and 35 genes, was generated by Cytoscape 3.9.1.

## Data Availability

FASTQ data have been deposited in the NCBI Gene Expression Omnibus and are accessible through GEO accession number GSE178889.

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
