# Peer review of "Patterns of Differentially Expressed circRNAs in Human Thymocytes"

_ncrna, 2022, doi:10.3390/ncrna8020026_

Round 1

Reviewer 1 Report

The paper by López-Nieva et al. brings relevant and original information on the role played by circRNAs in the differentiation of human thymocytes. The authors were able to show that the differential expression of circRNAs can be correlated with distinct stages of thymocyte development. As pointed out by the authors, these results pave the way for further investigations on how circRNA-miRNA-mRNA networks control the expression of genes involved in thymic T-cell differentiation.

The paper is well written and concise. The methodology employed for determining the differential expression of circRNAs in early immature, intermediate, and mature thymocytes is adequate and well described. The results and their interpretation and discussion are sound.

Nevertheless, there are a few key issues that should be considered by the authors in the present paper. Firstly, the authors do not provide the clinical and demographic data on the seven pediatric patients whose thymic explants were used as a source of thymocytes (or this reviewer was unable to find the data in the supplementary material). It is important to know how many patients with one month of age were included in this study, because the human thymus suffers a transient involution during the first month of postnatal life (see Varas et al. J Immunol. 2000 164(12):6260-7). It also important to know if these patients were karyotypically normal, since chromosomal alterations may cause thymic histological and functional abnormalities (Tomaru et al. Histopathology 2015, 67, 235–244). Additionally, is reasonable to suppose that these seven patients were submitted to cardiac surgery due to congenital heart disease (CHD). The thymuses of patients with CHD frequently present altered histomorphology and impaired cell cycle regulation (Ceyran et al. Int J Clin Exp Pathol 2015 8(7):8038-8047; Mestanova et al. Med Hypotheses. 2020 138:109599). Therefore, it would be necessary to consider these issues in the Introduction, Discussion, and Conclusion sections of the present paper.

Reviewer 2 Report

Dear authors,

The current manuscript dealing with the identification of differentially expressed circRNAs during thymocyte differentiation is interesting. The findings are intriguing about the role of non-coding circRNAs during differentiation process. The reviewer thinks that incorporation following suggestions would enhance the manuscript:

1) Introduction is very small and definitely needs to introduce the background related to this study and highlight the significance of this study. Introducing few related papers and physiological importance would gain readers attention.

2) Importantly, authors indicated in line 55/56 that thymocytes were isolated from thymuses of pediatric patients. However, the necessary methods regarding collection procedure or ethical permissions were not described in section 3. So, please add a section describing tissue collection from patients, and necessary IRB/ethical permissions.

3) Also, it is not clear whether these thymocytes at different stages were collected from thymus of one patient or pool of different tissues. If so, how many thymuses were collected and used in this study. please clarify

4) Did authors used any package other than DESeq2 for computation of diff. expressed circRNAs? Thus, authors can compare the sensitivity and identify high confident circRNAs.

5)  Fig.S1 (C), the qPCR validation plots can be moved to main paper and it is critical data validating rest of other findings. Addition of  table listing fold-change values from Seq data and qPCR for validated circRNAs would be appreciated.

6) Please plot a graph representing CLP variation data summarizing Table S1 in the main manuscript. CircTest will be a helpful tool for that purpose.

7) Please elaborate the results from mRNAs differentially expressed during thymocyte differentiation. (line 138/139). Two sentences is not indicating the rationale for such analysis and please discuss the findings and class/ontology analysis of those DEGs. how about a heatmap of mRNAs that were differentially expressed?

8) Please add a figure showing the circRNA-miRNA-mRNA networks in the main manuscript supporting Table-S4. If the network plot is big with many circRNAs, authors can show a represenataive network plot with few significant circRNAs.

9) Addition of a table listing circRNAs that were already reported for a role in differentiation process would really highlight the importance of diff. expressed circRNAs reported in this paper. 

10) some symbols such as >=/<= are confusing. Does it used for greater or equal to and less or equal to ? use appropriate symbols instead of using two symbols.

11) Little elaborated discussion of results with previously reported findings would be helpful. 

Best,

Round 2

Reviewer 2 Report

Dear Authors,

Thanks for providing the revised version of your manuscript after addressing all of the reviewer concerns. Overall, the revised manuscript is in good shape. I would recommend authors to incorporate following MINOR suggestions in the final version prior to publication.

1) In line 142 & 382, format 'ST1.ST2' to 'ST1-ST2' similar to that of other comparisons.

2) In Figure-4 (p-6), the y-axis labeling should be corrected from '0,5/1,5/2,5/3,5/4,5' to '0.5/1.5/2.5/3.5/4.5'.

3) Indicate 'n=3' in the Figure-4 legend

4) Correct '(SP1,SP3 and SP3)' to '(SP1, SP2, and SP3)' in line 178/179.

5) In line 182, end the sentence with '.'

6) Italicize genes names such as 'RGA2' in the manuscript

7) Figure-6 looks like there were more than 3 clusters as indicated in line 217. Please correct 

8) Cytoscape visualizing tool used in Figure-7 needs to be cited and indicated the respective methods section. Which tool was used to build the network in the Cytoscape?

9) correct 'm-RNA' to 'mRNA' in line 220 (figure-7 title).

10) In line 265, correct '...efficiency In order....' to '...efficiency. In order...' 

11) check the sentence in line 201/202. It started in a open bracket..may need to be closed.

12) Adding a 'Abbreviations' section would help the readers.

Best,
